A comparison of spinal and lower extremity biomechanics during maximal and sub-maximal deadlifts among strength-trained women

Gundersen Andreas H. andreasgundersen78@gmail.com
van den Tillaar Roland
Falch Hallvard
Larsen Stian
Department of Sports Science and Physical Education, Nord University , Levanger , Trøndelag , Norway
van den Hoek Daniel
Electronic publication date: 2025 Nov 3
Publication date: 2025
Volume: 13
Electronic Location ID: e20279
Received 2025 May 7; Accepted 2025 Oct 1
Copyright: ©2025 Gundersen et al.
Copyright year: 2025
Copyright holder: Gundersen et al.
License: This is an open access article distributed under the terms of the Creative Commons Attribution License, which permits unrestricted use, distribution, reproduction and adaptation in any medium and for any purpose provided that it is properly attributed. For attribution, the original author(s), title, publication source (PeerJ) and either DOI or URL of the article must be cited.
License URL: https://creativecommons.org/licenses/by/4.0/

Keywords: Resistance training, Biomechanics, Kinetics, Kinematics, Surface electromyography

Funding: The authors received no funding for this work.

==============================
Background

Previous studies have examined changes in biomechanical variables in response to different deadlift loads, yet, the effect of sub-maximal and maximal loads on potential deviation in lifting technique remain inadequately understood.

Methods

Therefore, this study compared barbell and joint kinematics, net joint moments (NJMs), and surface electromyography (sEMG) amplitude during 70%, 90%, and 100% of three-repetition maximum (3RM) load using statistical parametric mapping. Twelve strength-trained women (age: 23.18 ± 3.46 years, height: 166.72 ± 2.90 cm, body mass: 68.18 ± 7.67 kg) lifted 72.5 ± 9.3 kg, 93.9 ± 16.6 kg, and 102.9 ± 17.6 kg s at 70%, 90%, and 100% of 3RM deadlifts, respectively.

Results

The main findings revealed that the maximal load resulted in a significant increase in lower thoracic flexion angles and erector spinae sEMG amplitudes compared to the sub-maximal loads. Additionally, significantly lower hip NJMs were observed with a 70% load compared to 90% and 100% loads.

Conclusion

Therefore, increasing loads beyond 90% of 3RM might not be necessary if the goal is to train hip extensor strength through deadlifting. Deadlifting loads beyond 90% of 3RM may be achieved by increasing spinal flexion. This posture may allow strength-trained women to lift beyond the strength capacity of their hip extensors during the final repetition of a 3RM deadlift.

Introduction

The conventional deadlift (deadlift) is a multi-joint exercise frequently used to increase strength, power, and hypertrophy of the posterior chain muscles (Krajewski, LeFavi & Riemann, 2019; Stock & Thompson, 2014). The deadlift comprises one of the three disciplines used in competitive powerlifting, where the athlete is required to ascend to an erected position with both hips and knees locked (International Powerlifting Federation, 2024). Previous researchers have quantified biomechanical variables during deadlifts (Camara et al., 2016; Cholewicki, McGill & Norman, 1991; Escamilla et al., 2000; Martín-Fuentes, Oliva-Lozano & Muyor, 2020; McGuigan & Wilson, 1996; Swinton et al., 2011). Martín-Fuentes, Oliva-Lozano & Muyor (2020) revealed large surface electromyography (sEMG) amplitudes in spinal, hip, and knee extensors during the deadlift exercise. This is likely explained by the considerable external hip and spinal net joint moments (NJMs) induced by maximal deadlifts (Cholewicki, McGill & Norman, 1991; Escamilla et al., 2000). Regarding sub-maximal deadlifts, Swinton et al. (2011) compared joint angles of the hip, knee, and trunk between different sub-maximal deadlift loads (10%–80% one repetition maximum (RM)) and observed no significant difference in initial lifting posture or spatiotemporal extension patterns with increasing loads in male powerlifters. This indicates that the athletes did not change their kinematic self-organization strategies due to sub-maximal barbell loads, even with a 20% increase in hip net joint moments (NJMs) from 50% to 80% barbell load (Swinton et al., 2011). However, how barbell loads over >80% of one repetition maximum (1RM) impact kinematic self-organization strategies (e.g., spinal flexion) remains unstudied. From the perspective of powerlifting training, this gap is particularly significant, given maximal deadlift’s importance in competition and its status as the exercise where the heaviest loads are typically lifted.

Furthermore, in another fundamental multi-joint exercise, like the squat, self-organizational strategies, such as joint kinematic and kinetic changes, between sub-maximal and maximal loads have been extensively studied (Flanagan & Salem, 2008; Maddox & Bennett, 2021; Yavuz & Erdag, 2017). These studies observed unaltered knee NJMs but larger hip NJMs (Flanagan & Salem, 2008; Maddox & Bennett, 2021) and more forward lean (Yavuz & Erdag, 2017) with increasing loads. However, the extension of these findings to the deadlift is currently unknown, as comparative research between squat and deadlift has shown deviations in sEMG amplitude, kinematics, and kinetics of the lumbar and lower extremities (Ebben et al., 2009; Edington, 2017; Escamilla et al., 2000; Escamilla et al., 2001; Hales, Johnson & Johnson, 2009; Hamlyn, Behm & Young, 2007; Nuzzo et al., 2008). Hales, Johnson & Johnson (2009) observed that during maximal back squats, male powerlifters maintained a rigid lumbar lordosis, whereas maximal deadlift resulted in an abnormal lumbar curvature with a prominent thoracic kyphosis and rounded back posture. This difference may be attributed to the barbell’s position on the shoulders during squats, where excessive spinal flexion may contribute to a loss of balance, thereby rendering increased spinal flexion counterproductive. Conversely, during deadlifts, the barbell is hand-held. Thus, increasing spinal flexion during maximal deadlifts may enhance performance due to several biomechanical factors (Howe & Lehman, 2021). For example, an increased lumbar and thoracic flexion may provide a mechanically advantageous position by shortening the torso, thereby increasing the barbell’s proximity to the lumbar and hip joints, reducing the external moment arm and NJMs for all joints (Hales, 2010). Furthermore, spinal flexion aids in reducing peak hip flexion angles (Pinto, Beaudette & Brown, 2018), potentially preserving a better length-tension relationship and internal moment arms for the gluteus maximus and hamstrings (Németh & Ohlsén, 1985), which is critical given their role as primary movers during the deadlift (Martín-Fuentes, Oliva-Lozano & Muyor, 2020). Nevertheless, the potential deviations in lifting technique between sub-maximal and maximal loading conditions remain poorly understood. Thus, as strength training and strength development occur on a spectrum of heavy loading ranges (Schoenfeld et al., 2017), understanding the biomechanics of potential self-organizational strategies might enhance training specificity and deadlift performance as loading ranges change from sub-maximal to maximal.

Lastly, women are underrepresented in many of the deadlift studies referenced above, despite clear evidence that men and women differ in key anthropometric characteristics relevant to deadlift performance (Byl, Cole & Livingston, 2000; Herrington & Nester, 2004; Janssen et al., 2000; Kanehisa, Ikegawa & Fukunaga, 1994; Miller et al., 1993). For example, on average, women tend to exhibit a more pronounced Q-angle, which has been found inversely correlated with knee extensor strength (Byl, Cole & Livingston, 2000; Herrington & Nester, 2004). The scarcity of female samples in the literature complicates the interpretation of self-organization strategies in women during deadlift tasks due to gender differences in anthropometry. Consequently, more deadlift research focusing on strength trained women is needed as results from studies with primarily male samples might not be generalizable to women.

The aim of this study was to denote the effect of 70%, 90%, and 100% of 3RM deadlift loads on NJMs of the lower limbs, as well as the kinematics and sEMG amplitudes of the spine and lower limbs among strength-trained women. Given the frequent observation of increased spinal flexion during maximal deadlifts (Hales, 2010; Hales, Johnson & Johnson, 2009), and distinct self-organization patterns observed between sub-maximal and maximal loads in the squat exercise (Larsen et al., 2022; Maddox & Bennett, 2021), we hypothesized that: (1) spinal flexion angles would increase progressively with load, being smallest at 70% 3RM and largest at 100% 3RM; and (2) hip NJMs would follow a similar trend, with the smallest NJMs at 70% 3RM, followed by 90%, and largest at 100% 3RM.

Materials & Methods

Experimental design

To address the underexplored effect of sub-maximal and maximal deadlift loads, a within-subject crossover design was employed to quantify kinematics, NJMs, and sEMG amplitude during the final repetition of a three repetition maximum (3RM). Each participant participated in a familiarisation session followed by an experimental test session. During familiarisation, participants received detailed feedback from two experienced powerlifting coaches to optimise their performance during the test session. Seven days of rest was mandated between the sessions to prevent fatigue. The dependent variables in this study included joint angles, NJMs, barbell velocity, and sEMG amplitude, while the independent variables were the different deadlift loads.

Participants

We recruited twelve strength-trained women (age: 23.18 ± 3.46 years, height: 166.72 ± 2.90 cm, body mass: 68.18 ± 7.67 kg). Four participants were competitive powerlifters and seven trained deadlifts regularly for non-competitive purposes. Inclusion criteria required participants to lift at least 1.5 times their body mass for a 1RM with hips fully extended and to be free from any injury that could affect their performance. Participants were advised to avoid any form of training, intense physical activity, or alcohol consumption 48 h before both sessions. Informed written consent was obtained from all participants. The study was approved by the Norwegian Agency for Shared Services in Education and Research (project number: 404365) and adhered to the latest Helsinki Declaration guidelines.

Sample size rationale

An adequate sample size of n = 12 was determined based on Falch et al. (2024) to find significant and medium-sized effects in joint kinematics and sEMG amplitude with an actual power of 0.80. Although 1D analyses, such as those conducted in the present study, generally require larger sample sizes to achieve sufficient power (Robinson, Vanrenterghem & Pataky, 2021), practical constraints limited recruitment beyond 12 participants. Therefore, to further increase our study’s robustness, we implemented a within-subject design. As a result, our study included twelve women with extensive strength training experience.

Procedures

The familiarisation session commenced with measuring participants’ body height and horizontal acromion length (right to left) using a measuring tape. Following this, body mass was recorded using a Tanita scale (MC-780MA, Riga, Latvia). Additionally, each participant filled out a questionnaire detailing their prior 1RM. A stance width equal to 1.0 times the acromion length was marked with tape and used for both sessions, with participants instructed to keep the medial part of the calcaneus on the tape for all repetitions. The experimental and familiarisation sessions followed similar protocols, starting with a general standardised warm-up (Larsen, Kristiansen & van den Tillaar, 2021) consisting of three sets of 10 repetitions with an unloaded Olympic barbell (Rogue, Ohio power bar). Thereafter, in the familiarisation, participants progressively added weight (typically 2.5–10 kg when approaching maximal), culminating in performing three repetitions at 100% 3RM. To enhance accuracy, measurements of mean barbell ascent velocity were conducted, and the value for the maximal 3RM in familiarization was saved for use in the experimental test. The 100% 3RM load was multiplied by 0.7 and 0.9 to estimate 70% and 90% of the 3RM for the experimental test session. During experimental testing, three repetitions of the pre-determined 70%, 90%, and 100% 3RM were executed in a randomised order using an online randomiser (https://www.random.org). To minimize the impact of daily fluctuations in strength and readiness on the 100% 3RM, the load was adjusted up or down by 1–10 kg based on the proximity to the mean ascent barbell velocity recorded during familiarization, or failure to reach accepted lock-out. The highest completed load with accepted technique was defined as 100% 3RM and selected for analysis. During testing, participants rested for 180 s between warm-up sets and 240 s between experimental sets (De Salles et al., 2009). Importantly, our goal was to compare deadlift variables without the constraints of powerlifting rules to observe how the body self-organises under different loading conditions. Consequently, the lockout position was deemed acceptable when the hip was fully extended, regardless of the knee flexion angle.

Measurements

Three-dimensional kinematic data were collected using Qualisys software (Qualisys, Gothenburg, Sweden) and eight motion capture cameras (Qualisys series 4 and 5+) sampling at 500 Hz to track reflective markers. Two Kistler force plates (type 9260AA6, Winterthur, Switzerland) sampling at 2,380 Hz were integrated with the Qualisys system provided ground reaction force data. Thirty-four markers were placed on anatomical landmarks, including the bilateral barbell ends, acromion, 7th and 10th thoracic vertebrae, 1st, 3rd, and 5th lumbar vertebrae, anterior superior iliac spine (ASIS), posterior superior iliac spine (PSIS), bilateral condyles of the knee, lateral and medial malleolus, sternum, tuber calcanei, and 1st and 5th proximal phalanx. Lumbar cluster markers were also placed bilateral of the lumbar vertebrae markers. The 12th thoracic vertebra was created as a landmark at the 10th thoracic vertebra with a 66% offset towards the 1st lumbar vertebra. After recommendations from the developer (C-motion, Germantown, MD, USA), using static trials, we computed planar angles in the YZ projection plane through three-point measurements involving the heel centre, centre, and ankle centre. These measurements were then incorporated into the ankle flexion angle to account for the ankle offset. sEMG data were collected with Trigno Avanti sensors (Delsys, Natick, MA, USA) at 1,111 Hz on five muscles (e.g., erector spinae iliocostalis, gluteus maximus, gluteus medius, semitendinosus, and vastus lateralis) following SENIAM guidelines (Hermens et al., 2000). The Trigno Avanti sensors were then coupled with the Qualisys system, and sEMG data was synchronized with motion capture data via the Qualisys software. Before attachment, the participants were shaved and cleansed with alcohol to avoid noise from hair and dead skin.

Data analysis

Kinematic data were gap-filled and cut to only include the concentric phase (defined between the moment the lifter has initiated the pull and lock-out) in Qualisys software. For segment building and analysis, all data were transported via C3D files to Visual3D (C-motion, Germantown, MD, USA) and filtered with a 6 Hz low-pass Butterworth filter to enhance accuracy, given the slow nature of the analysed movements. To further increase accuracy of inverse dynamics calculations, kinematic and force data were filtered with the same filter (Kristianslund, Krosshaug & Van Den Bogert, 2012; Mai & Willwacher, 2019). A Visual 3D composite pelvis was created using the bilateral ASIS and PSIS markers, and hip joint centre was determined using a built-in regression in equation function. Joint angles were calculated using direct kinematics and defined as the distal segment relative to the proximal segment in a X-Y-Z Cardan sequence. To measure lower thoracic and upper lumbar flexion, and reduce interference from soft tissue artefacts (Xi et al., 2022), a multi-segmental spine model was assessed (Papi, Bull & McGregor, 2019). The model comprised three segments: lower thoracic (T7-T12), upper lumbar (L1-L3), and lower lumbar (L3-L5). The lower thoracic angle was defined as the angle between the lower thoracic and upper lumbar segments, while the upper lumbar angle was defined as the angle between the upper and lower lumbar segments (Papi, Bull & McGregor, 2019). Three-dimensional NJMs for the hip, knee, and ankle were computed within a resolute coordinate system using the right-hand rule convention through inverse dynamics. Anthropometric data for inverse dynamics calculations were based on marker-data, participant body mass and height, and the Visual 3D integration of the geometric shape of segments. All joint moments representing external NJMs are reported as means and standard deviations relative to the resolute coordinate system of the distal segment. NJMs for the hip, knee, and ankle were summed bilaterally. Sagittal plane NJMs represent flexion/extension moments and were normalised to participants’ body mass and expressed in Nm/kg (Larsen, Kristiansen & van den Tillaar, 2021). Barbell velocity was calculated in Visual3D, using the movement velocity (m/s) of the barbell center of mass along the Z-axis relative to the laboratory. The raw sEMG signals were filtered using high-pass and low-pass filters set at 20 Hz and 500 Hz, respectively. Following this, the sEMG signals were fully wave rectified, and the mean root mean square values were calculated.

Statistical analysis

Data processing and statistical analysis were executed in MATLAB R2024A (The MathWorks, Natick, MA, USA). All deadlift parameters were first time-normalised from 0%–100% of the concentric phase using an interpolation function. Thereby, statistical analysis was performed through statistical parametric mapping (SPM) using a custom-made MATLAB application developed by A.H.G. The application was built around the open-source SPM tool package spm1d 0.4 (Pataky, 2012). A repeated measures analysis of variance (ANOVA) based on random field theory was conducted to evaluate the impact of different deadlift loads (e.g., 70%, 90% and 100%) on kinematics, kinetics, and sEMG amplitude. Significant regions were further analysed using two-tailed paired t-tests with a Bonferroni correction. Statistical non-parametric mapping results were assessed for instances when the spm1d normality test reached significance. Statistical significance was established at an alpha level of p < 0.05.

Results

Kinematics

The SPM ANOVA revealed significant main effects of load on barbell velocity, commencing between 15%–67% of the normalized concentric phase (0% and 100% represents the first and last frames of the concentric phase, respectively, Fig. 1). SPM post hoc analysis revealed significantly higher barbell velocities with the 70% load compared to the 100% and 90% loads between 27%–63% and 39%–51% in the concentric phase, respectively. Additionally, the 90% load induced significantly higher barbell velocities between 31%–64% of the concentric phase compared to the 100% load. Moreover, a significant main effect was discovered between 51%–70% of the concentric phase when comparing knee flexion angles. Post hoc SPM analysis revealed significantly larger knee flexion angles with the 100% load compared to the 70% load, occurring between 53%–65% of the concentric phase. Lastly, the SPM ANOVA analysis revealed a significant main effect on lower thoracic flexion angles between 14% and 66% of the concentric phase. More specifically, significantly larger lower thoracic flexion angles were observed with the 100% load compared to the 70% load between 38%–44% of the concentric phase.

Figure 1 SPM figure of kinematics.

The top row displays mean (±SD) barbell velocity (m/s), hip flexion, knee flexion, ankle plantar flexion, lower thoracic flexion, and upper lumbar flexion angles time normalised from 0%–100% with the 100% (solid lines), 90% (dashed lines), and 70% (dotted lines) 3RM deadlift loads during the concentric phase. The second row displays SPM ANOVA curves, with a grey-shaded area representing significant main effects when surpassing the significance line (red dotted line, p ≤ 0.05). The bottom three rows show SPM paired t-test curves with a Bonferroni correction.

Net joint moments

When comparing NJMs, the SPM ANOVA analysis revealed a significant effect of load only on hip extension NJMs (Fig. 2). More specifically, significant hip extension NJM main effects between 0%–28% and 62%–100% of the concentric phase were observed. SPM post hoc tests revealed that significantly smaller hip extension NJM occurred with the 70% load compared to the 100% (between 8%–14% and 80%–92%) and 90% loads (between 20%–28% and 76%–100%) during the concentric phase.

Figure 2 SPM figure of NJMs.

The top row displays mean (±SD) hip, knee, and ankle NJMs (Nm/kg) time normalised from 0%–100% with the 100% (solid lines), 90% (dashed lines), and 70% (dotted lines) 3RM deadlift loads during the concentric phase. The second row displays SPM ANOVA curves, with a grey-shaded area representing significant main effects when surpassing the significance line (red dotted line, p ≤ 0.05). The bottom three rows show SPM paired t-test curves with a Bonferroni correction.

Surface electromyographic amplitude

When comparing sEMG amplitude, the SPM ANOVA analysis revealed a significant main effect of load in the gluteus medius between 62%–67% of the concentric phase (Fig. 3). SPM post hoc analysis revealed a significantly larger sEMG amplitude with the 100% load compared to the 70% load between 62%–67% of the concentric phase. Moreover, the SPM ANOVA revealed a significant main effect of load in the erector spinae, commencing between 50%–63% of the concentric phase. SPM post hoc tests revealed a significantly larger sEMG amplitude in the erector spinae with the 100% load compared to the 90% and 70% loads between 54%–55% and 51%–61%, respectively.

Figure 3 SPM figure of EMG.

The top row displays the mean (±SD) sEMG amplitude for the different muscles’ time normalised from 0%–100% with the 100% (solid lines), 90% (dashed lines), and 70% (dotted lines) 3RM deadlift loads during the concentric phase. The second row displays SPM ANOVA curves, with a grey-shaded area representing significant main effects when surpassing the significance line (red dotted line, p ≤ 0.05). The bottom three rows show SPM paired t-test curves with a Bonferroni correction.

Discussion

This study examined the effect of 100%, 90%, and 70% of 3RM deadlift loads on NJMs of the lower limbs, along with the kinematics and sEMG amplitudes of the spine and lower limbs during the final repetition of a three-repetition set in strength-trained women. The main findings supported our initial hypothesis, as the maximal load resulted in significantly greater lower thoracic flexion and higher erector spinae sEMG amplitudes than the sub-maximal loads. Additionally, hip NJMs increased significantly from the 70% load to the 90% and 100% loads, but not from 90% to 100%.

To our knowledge, this is the first study to compare kinematics between sub-maximal and maximal deadlift loads, revealing greater lower thoracic flexion angles with the maximal load. Interestingly, no difference in spinal flexion was observed between the sub-maximal loads (Figs. 1 and 4). This may be attributed to the participants lifting more than their hip extensor strength with the maximal load, prompting them to flex the spine for a biomechanically advantageous position (Howe & Lehman, 2021). Similarly, Hales, Johnson & Johnson (2009) observed substantial spine flexion among powerlifters during maximal deadlifts, later interpreted to enhance performance by shortening the torso, thus increasing the barbell’s proximity to the hip and spinal joints (Hales, 2010). This position also allows for a more extended hip, potentially increasing the internal moment arm of the gluteus maximus and hamstrings (Németh & Ohlsén, 1985), which is critical given their role as primary movers during the deadlift (Martín-Fuentes, Oliva-Lozano & Muyor, 2020). As such, the participants in the current study might have employed a self-organization strategy of increasing spinal flexion to manage the increased load. Also, our findings indicate a gradual reduction in barbell velocity from the lightest to the heaviest load. This is in line with previous literature (Morán-Navarro et al., 2021) and very logical considering the increased barbell inertia and leftward shift of the force-velocity profile with increasing barbell loads.

Figure 4 Showing spine flexion.

From left to right: 70%, 90%, and 100% 3RM, illustrating flexion in the multi-segmental spine model from a representative subject during the concentric phase. From top to bottom: lower thoracic segment (red), upper lumbar segment (green), and lower lumbar segment (blue).

Moreover, hip NJMs increased from the 70% load to the 90% and 100% loads. This is comparable to a previous study who reported a 20% increase in hip NJM from 50% to 80% of 1RM deadlifts in male powerlifters (Swinton et al., 2011). Similarly, increasing hip NJMs have been observed with increasing loads in the squat exercise (Flanagan & Salem, 2008; Larsen et al., 2022; Maddox & Bennett, 2021). However, no increase in hip NJMs beyond the 90% load was observed, which contrasts the squat research (Larsen et al., 2022; Maddox & Bennett, 2021). This may suggest different self-organization strategies between the squat and deadlift when transitioning from sub-maximal to maximal loads. It is postulated that this observation was due to the handheld barbell, which might allow greater spinal flexion. The increased spinal flexion might enable participants to lift beyond their hip extensor strategy, potentially explaining why powerlifters typically lift more in the deadlift than in the squat.

Lastly, higher erector spinae sEMG amplitude with the maximal load was observed compared to the submaximal loads. This was attributed to the participants lifting with increased spinal flexion at maximal load, potentially enhancing the force production capacity of the erector spinae (Dolan, Mannion & Adams, 1994; Holder, 2013; Piazzesi et al., 2002; Purslow, 1989). Additionally, increased spinal flexion and barbell load means a larger extension effort is required to extend the spine and complete the lockout. As such, the erector spinae may have contributed more to the maximal load, consequently increasing sEMG amplitude. However, this cannot be concluded by EMG data alone, and further analysis using musculoskeletal modelling of the spine should be conducted to draw more robust conclusions.

Practical applications

Our study offers valuable insights for strength and conditioning practitioners, especially those working with strength-trained women. Based on our findings, it may not be necessary to increase loads beyond 90% of 3RM to maximize hip extensor strength through deadlifting. Instead, lifting loads beyond 90% appears to be achieved by increasing spine flexion.

Limitations

This study has several limitations that must be addressed. First, statistical parametric mapping based on random field theory was used. This methodology did make it difficult to estimate effect sizes. Secondly, the athletes in our study were not required to fully extend their knees during the lockout phase. As a result, our findings may not be directly comparable to previous research or to athletes who adhere to the standards set by the International Powerlifting Federation (International Powerlifting Federation, 2024). Time of day, hydration, and ergogenic aid use were not controlled for in this study. Although a within-subject design was employed, these factors may have contributed to increased variability in the data. Furthermore, we exclusively reported NJMs calculated via inverse dynamics, which do not account for individual muscle forces (Vigotsky et al., 2019). Also, the study design did not include a range of motion normalisation test for the spine kinematics, complicating the interpretation of results for future research (Edington, 2017). In addition, our study used a skin-marker based multi-segmental spine model, which is susceptible to measurement errors of approximately 10 mm due to soft tissue artefacts (Xi et al., 2022; Zemp et al., 2014). While these implications are likely minimized in our study due to the within-subject design, future studies should be cautious when directly comparing their results to ours, as soft tissue artefact may introduce variability. Moreover, sEMG amplitude was not normalised, which might increase noise, and the risk of type-1 error. However, as it was a within-subject design, we expect that this noise would be similar within each subject over the different loads. Lastly, our model did not allow for the estimation of spinal NJMs. Thus, future studies should include spinal NJMs and musculoskeletal modelling to better understand spinal parameters with increasing deadlift loads.

Conclusions

Based on our findings, increasing loads beyond 90% of 3RM might not be necessary if the goal is to train hip extensor strength through deadlifting. Lifting loads beyond 90% of 3RM may be achieved by a self-organizational strategy involving increased spinal flexion, probably to put the hip extensors in a biomechanically advantageous position and reduce external moment arms. This self-organization strategy may allow strength-trained women to lift beyond the strength capacity of their hip extensors during the final repetition of a 3RM deadlift.

Supplemental Information

Supplemental Information 1 Raw data for kinematics, NJMs, and sEMG

The authors would like to thank all participants for their contributions to this study. Finally, the authors acknowledge the use of ChatGPT (OpenAI, GPT-4) for language editing support, as none of the authors are native English speakers.

Additional Information and Declarations

Competing Interests

Author Contributions

Human Ethics

Data Availability

The authors declare there are no competing interests.

Andreas H. Gundersen analyzed the data, prepared figures and/or tables, authored or reviewed drafts of the article, and approved the final draft.

Roland van den Tillaar conceived and designed the experiments, authored or reviewed drafts of the article, and approved the final draft.

Hallvard Falch conceived and designed the experiments, performed the experiments, authored or reviewed drafts of the article, and approved the final draft.

Stian Larsen conceived and designed the experiments, performed the experiments, analyzed the data, authored or reviewed drafts of the article, and approved the final draft.

The following information was supplied relating to ethical approvals (i.e., approving body and any reference numbers):

The study was approved by the Norwegian Agency for Shared Services in Education and Research, 404365.

The following information was supplied regarding data availability:

The raw data is available in the Supplementary File.

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
