# Peer review of "A comparison of spinal and lower extremity biomechanics during maximal and sub-maximal deadlifts among strength-trained women"

_PeerJ, doi:10.7717/peerj.20279_

## Round 0.1 · original submission · Major Revisions

The manuscript requires development of clarity about referenced studies (i.e., including the study sample descriptors) and statistical testing results. Please see specific areas to address within the reviewer comments and provide a point-by-point response to their feedback.

Reviewer 1 ·

Basic reporting

Title & Abstract
Title:
The title is sound and represents the study adequately.
Abstract:
The Abstract is well written with adequate structure, yet I recommend that the results here include p values and some actual data to be more clear.

Introduction
The review of prior literature is sound here, but please add the study sample in any study mentioned e.g. your sample is comprised of women, who have a different body type and muscle mass than men, so data acquired in men regarding the deadlift cannot be applied to women.
Ln 42 use of the word ‘understudied’ reveals that some studies do exist, so I recommend that new text be added here to briefly articulate key findings from these prior works, and what remains unknown.
This study may add to your background and data evaluation, although it was performed in men. https://pubmed.ncbi.nlm.nih.gov/24276311/.
Other than these issues, this section is well written and effectively presents the study background and need for this study.

Figures & Tables
Other than Fig1, all figures included are clear and understandable. In my view, there is just too much information in Fig1 which makes its content really challenging to digest. For example, there are 30 different graphs included here, yet the text reporting these data is quite brief.

Experimental design

Material and Methods
Was pre-exercise food intake considered? Hydration? Use of caffeine which is a known ergogenic aid? Was time of day maintained within participants?
The composition of this sample is somewhat mixed e.g. athletic and some who typically do deadlifts; does this disparity impact your project?
Ln 100 This explanation of sample size is inadequate; I recommend that you do a post hoc power analysis to determine if your N=12 is sufficient to detect a meaningful change in your main outcome (joint moment). Also, it may be useful for you to report Ns used in prior similar studies.
Ln 116 for greater clarity, please add text to state how much weight was typically increased across these sets.
Please briefly explain why these specific loads were used in the present study. Also, please explain why data were only acquired from the last repetition of exercise when fatigue may be more pronounced.
Ln 125 there is potential for CNS fatigue due to heavy resistance exercise; please justify the 3 and 4 min recovery provided between sets and exercises.
There is adequate detail included in this section, especially text related to data acquisition and analysis. Yet, the study timeframe is not listed and the estimation of power is weak.

Validity of the findings

Results
Ln 192 202 this text would be more clear if p values for all analyses were listed here in the Results section.
The data presented here as % (e.g. between 27% and 63% and 39 and 51% in the concentric phase) need greater explanation, as it is not evident what these values represent.
Overall, this section is well organized and clearly written. The results complement the methods. This paper makes a contribution to the data in this field, and the results seem plausible.

Discussion
Ln 238 please clarify the word ‘indorsed’ here; thank you.
Ln 248 do you mean 50 to 80 %1RM? Please clarify the text here.
Ln 255 requires a reference citation here to validate this finding.
This text does a solid job of restating the study aim, listing the main results, and then evaluating them based on prevailing literature. Areas of future study as well as study limitations and its application to the field are also addressed. Overall, this section is well written.

Conclusion
The Conclusion is well articulated and effectively denotes the take home message of this manuscript. No modification is needed in my opinion.

·

Basic reporting

The authors have provided a clear, detailed commentary on their investigation. They clearly introduce the importance and the gap it fills with reference to relevant literature to support. The structure conforms to discipline norms, with relevant figures which are well labelled and described. Raw data has been supplied.

Experimental design

The study is original research within the scope of the journal, with a clearly defined research question. The relevance and importance of the research question is well described as well as the gap in knowledge the authors were addressing. The investigation was performed to a high standard with methods for the most part provided in enough detail to replicate. There are a couple of parts in the methods that could use more clarity, these are outlined below.

Validity of the findings

Data have been provided with clearly outlined statistical methods and within subject design to act as own controls. Conclusions are well supported by the results, and clearly answer the research question.

Additional comments

I commend the research team on a great and interesting piece of research. In particular it is nice to see strength research on a female cohort. I have included some specific comments below to address.
Introduction
Line 30 – delete “from a powerlifting standpoint”
Line 61 – where you say “greater length-tension relationship” do you mean increased? Or more optimal?
Line 75 – full stop is needed at the end of the sentence.

Methods
Lines 101-105 – The section on the sample size justification is unclear. Was there a power calculation? Either a priori or post hoc? This paragraph doesn’t really tell the reader anything.
Line 129 – Surely the knee angle would also impact how the body self organises?
Line 129 – needs a full stop at the end of the sentence.
Lines 142-143 – A figure to illustrate the computed angles at the ankle would be useful.
Lines 152-153 – How were the onset of movement and complete lockout identified in the data?
Line 157 – define V3D, ASIS, and PSIS.

Results
Line 195 – The 90% load induced higher barbell velocities than what?
Lines 196-199 – the significant results in knee flexion are not included in the discussion. I suggest some discussion be added of this.

Discussion
Practical applications – font size changes for this section.
Line 270 - I'm not sure you can say it is primarily achieved by the spinal extensors, just that there is a greater contribution from the spinal extensors.
Line 280 – edit “tests” to “test”
Line 281 – delete “hard”

---

## Round 0.2 · Minor Revisions

Thank you for addressing the previous comments of the reviewers. The reviewers have provided further comments to improve the clarity of your work. Please revidse your manuscript and with a point by point response to the reviewer comments.

Reviewer 1 ·

Basic reporting

Title & Abstract
They effectively represent the content contained in this article, but data need to be inserted here to be more transparent; reporting actual data is highly recommended in any Abstract to better summarize the primary study findings. The conclusion should not speculate excessively, and only evaluate findings acquired in this study specifically mentioned in the Abstract.
Background revise to; Previous studies have examined changes in biomechanical variables in response to different deadlift loads…yet, the effect of potential deviation in…
The methods here is repetitive as 3RM and the corresponding 3 loads listed are denoted here twice, which needs improvement. I recommend that their notation be removed in the last sentence included here. For more detail, briefly explain what kinematic variables were assessed.
Results presented here require actual p values and some actual presentation of data as mean SD—without this, this Abstract is still unclear and does not effectively denote the main study findings.
Self-organization should be removed here, as no text is presented here to make this term ‘fit.’ My recommendation is to remove these terms from this conclusion.

Introduction
Ln 40 46 participants in these studies are not noted, so it is not clear if these are women, athletes, etc. Please list these populations here and in all text when denoting findings from prior studies.
Ln 48 please define the term self-organization here to be more clear.
Ln 62 It is not clear if these are significant changes or differences? Here and in this text, please include the word ‘significant’ where it applies to adhere to typical scientific language.
Ln 69 please use the word ‘strength training’ here as this is better understood, rather than athletic training.
Ln 84 please revise the aim statement to denote the effect of different deadlift loads on NMJ in strength trained women. The word effect better denotes the study design.
Overall, the background is adequate with solid study rationale and novelty and a clear aim and hypothesis.


Figures & Tables
The Figures are adequate and complemented by the legends provided with the figures.

Experimental design

Material and Methods
There is adequate detail here, yet the study timeframe should be listed here. The methods so fit with the study’s primary aims. Below are some issues which I ask the Authors to consider.
Ln 96 please revise this to denote that this study used a within subject’s crossover design. The word ‘relationship’ is not accurate as you are looking for effects of different loads, not relationships.
Ln 106 please change to We recruited ... or participants were===’the study recruited’ is personification which is inadequate text.
Ln 117 this power estimation is appreciated, yet which outcome is this designed to find effects for? Please add this here.
Data normality and sphericity are key assumptions of RM ANOVA; please denote if these were tested and met.
The Authors downplay the effect of time of day, hydration, and possible ergogenic aid use on their responses, since this is a within subjects’ design. This is true, yet data have greater variability if these factors are not controlled or considered. Please cite these as study limitations and I highly recommend you to account for these important factors in subsequent work in your lab.
This section is well organized and very thorough and clear.
The authors’ response that the last of 3 points is a maximal repetition needs validation, as my experience performing resistance training and reading this literature is that the first repetition of a given set may be ‘maximal’, with the least incidence of fatigue. If any potential differences in force and/or EMG are minimal between the first and 3rd repetition, then state this and it is best if this is backed with empirical results.

Validity of the findings

Results
Ln 211 222 report the results clearly, yet no p values are denoted throughout which I recommend should be placed in (). Moreover, since you are using inferential statistics, the word ‘significant’ needs to be inserted where applicable i.e. when terms such as higher are used, lower, larger, etc. when comparing various scores with post hoc analyses.
Please also do this for the subsequent 2 paragraphs of Results; insert proper p values and use the word ‘significant’ where it is applicable to complement the text.
These results incrementally advance the field although the small sample limits the study impact.
In my view, these data seem plausible.

Discussion
Ln 246 Please revise to This study examined the effect of different deadlift loads on... (list your outcomes). This is a more precise way to denote the study aim.
Ln 258 please replace ‘large’ with substantial, a more appropriate term.
Ln 265 267 also align with the Force-velocity profile and this should be denoted here as an alternative explanation.
Ln 269 272 change the reference structure, so please revise this; moreover, denote the participants in these studies, which should be athletic women for a better comparison amongst studies.
Ln 286 please replace ‘concise’ with robust or similar word.
Other limitations not mentioned include the inability to apply these data to any other sample, use of barbell deadlift rather than RDL, hip thrust, etc. and no measure of body composition (muscle mass) which may mediate some of the trends revealed in your data.
Overall, the findings correlate with the results, which are relevant to the broader area of research of kinematics of strength training. The text effectively explains the data and compares these findings to prior results.



Conclusion
The Conclusion is well written and along with the Practical Applications, effectively states the take home message of this study.

·

Basic reporting

Reference formatting needs to be checked throughout- there are a number of errors and reference style changes partway through the discussion. Please check this for consistency.

Experimental design

No comment

Validity of the findings

No comments

---

## Round 0.3 · accepted · Accept

Thank you for your diligent efforts to address the reviewer comments. Your article is now suitable for acceptance by the journal. Congratulations.

Reviewer 1 ·

Basic reporting

Title & Abstract
The title is very clear in summarizing the main elements of this study. The revised abstract is well written and needs no revision, as it effectively captures the study background-aim, methods, results, and conclusions.

Introduction
The study background is very well constructed and makes a strong case for the need for this study and its novelty. Please explain your decision or reasoning.

Figures & Tables
These are adequate as presented in the paper.

Experimental design

Material and Methods
Ln 99 Please revise to each participant ‘completed’ a… there are too many uses of the word ‘participant’ here which is a little awkward.
Please refer to session 2 as the ‘experimental session’ and remove the word test, which is vague.
Ln 136 please revise text ‘…. to value for the maximal 3RM was used’…
Other than these minor comments, I have no issues with this section of text which is well organized, thorough, and easy to follow.
Statistical power has been included in the text, and the statistics follow from the study design.

Validity of the findings

Results
The presentation of the Results is very clear and easy to follow. These results do add to this body of knowledge and do seem plausible.

Discussion
Ln 260 please revise text to Hales et al. (2009)…
The text does an effective job of restating main study findings and interpreting them according to literature. The study limitations presented are sound.

Conclusion
The revisions are sufficient. Ln 321 Please revise text to ‘Our data show that increasing loads..’

·

Basic reporting

Some good improvement of the manuscript throughout the review process. I'm happy with the changes made.

Experimental design

No comment

Validity of the findings

No comment